# Metagenomics-Based Analysis of Candidate Lactate Utilizers from the Rumen of Beef Cattle

**DOI:** 10.3390/microorganisms11030658

**Published:** 2023-03-04

**Authors:** Venkata Vinay Kumar Bandarupalli, Benoit St-Pierre

**Affiliations:** 1Department of Animal Science, South Dakota State University, Animal Science Complex, Box 2170, Brookings, SD 57007, USA; 2GenMark Diagnostics, 5964 La Place Ct, Carlsbad, CA 92008, USA

**Keywords:** rumen, microbiome, 16S rRNA gene, lactate, acidosis

## Abstract

In ruminant livestock production, ruminal acidosis is an unintended consequence of the elevated dietary intake of starch-rich feedstuffs. The transition from a state of subacute acidosis (SARA) to acute acidosis is due in large part to the accumulation of lactate in the rumen, which is a consequence of the inability of lactate utilizers to compensate for the increased production of lactate. In this report, we present the 16S rRNA gene-based identification of two bacterial operational taxonomic units (OTUs), Bt-01708_Bf (89.0% identical to *Butyrivibrio fibrisolvens*) and Bt-01899_Ap (95.3% identical to *Anaerococcus prevotii*), that were enriched from rumen fluid cultures in which only lactate was provided as an exogenous substrate. Analyses of in-silico-predicted proteomes from metagenomics-assembled contigs assigned to these candidate ruminal bacterial species (Bt-01708_Bf: 1270 annotated coding sequences, 1365 hypothetical coding sequences; Bt-01899_Ap: 871 annotated coding sequences, 1343 hypothetical coding sequences) revealed genes encoding lactate dehydrogenase, a putative lactate transporter, as well as pathways for the production of short chain fatty acids (formate, acetate and butyrate) and for the synthesis of glycogen. In contrast to these shared functions, each OTU also exhibited distinct features, such as the potential for the utilization of a diversified set of small molecules as substrates (Bt-01708_Bf: malate, quinate, taurine and polyamines) or for the utilization of starch (Bt-01899_Ap: alpha-amylase enzymes). Together, these results will contribute to the continued characterization of ruminal bacterial species that can metabolize lactate into distinct subgroups based on other metabolic capabilities.

## 1. Introduction

As herbivores, ruminants have the ability to transform human-inedible plant biomass into products that humans can consume or utilize [1]. This has not only greatly benefited humanity throughout its history, but it is also predicted to continue to serve this purpose as we strive for improved environmental sustainability of food production [2,3]. Because genes encoding enzymes that can efficiently digest plant fibers are not present in the genomes of mammals or other animals, ruminants rely on the metabolic activities of gut symbiotic microorganisms to break down and ferment plant structural polysaccharides into nutrients that they can absorb and assimilate [4]. In ruminants, the fermentation of feed takes place in the rumen, the largest segment of their compartmentalized stomach, which is consequently the main site where microbial symbionts reside in these herbivores [5]. Rumen microorganisms consist of bacterial, archaeal, protozoal, and fungal species, and these assemble into complex microbial ecosystems as a result of trophic relationships and intricate functional networks that form amongst symbionts [6,7]. Rumen microbial communities are responsible for fermenting feedstuffs, which provide nutrients in the form of microbial proteins and short chain fatty acids (SCFAs) for their host [8]. Since approximately 70% of the energy utilized by a ruminant is from SCFAs, ruminal microbial communities are crucial contributors to ruminant productivity [9].

A common practice in intensive feedlot and dairy cattle production systems is to feed high levels of grain or concentrate. As these ingredients contain higher levels of compounds such as starch, which are more easily digestible than plant fibers, they can be metabolized more rapidly, allowing improved animal production by providing higher quantities of nutrients that can be utilized by the host. However, since increased digestibility rates result in a greater accumulation of ruminal SCFAs, feeding easily fermentable diets can cause the ruminal pH to become too acidic [10]. This can result in subacute acidosis (SARA), a common digestive disorder clinically defined as when ruminal pH values are between 5.0 and 5.6 for an extended period of time [11,12]. This condition can be further exacerbated by the accumulation of lactate, a more potent organic acid that can cause acute acidosis as pH values drop below 5.0 [13].

In animal nutritional models, experimental induction of SARA has been associated with reduced ruminal bacterial diversity and richness, as well as changes in ruminal microbial community composition [10,14,15,16,17,18,19,20,21,22]. Consistent with these effects of acidosis on microbial community structure, investigations using metagenomics and metatranscriptomics have revealed that a number of metabolic functions are affected by acidosis, including biofilm formation as well as carbohydrate, amino acid and energy pathways [23,24]. A number of studies have identified bacterial species such as *Streptococcus bovis* and members of the genus *Lactobacillus* as dominant lactic-acid producers, while other species, such as *Megasphaera elsdenii* and *Selenomonas ruminantium* subsp. *lactilytica*, have the ability to metabolize lactate to counter this effect [7,14,24,25,26,27].

While higher inclusion of easily fermentable feedstuffs leads to a higher representation of both lactate producers and utilizers, the metabolic capacity of lactate utilizers is not sufficient to compensate for the levels of lactate being produced once lactate levels reach a critical point [28,29]. This would explain why gradual adaptation from a roughage to a concentrate diet remains a common and recommended practice, despite the lower gains and higher feed expenses that come from this practice. Aside from well-characterized lactate utilizers such as *M. elsdenii* and *S. ruminantium*, our knowledge of ruminal lactate utilizers remains limited. Considering that the vast majority of ruminal microorganisms remain uncharacterized [30,31], it would be reasonable to assume that other lactic acid-utilizing bacteria remain to be identified in this environment. Thus, in an effort to gain more insights into lactate metabolism in the rumen, this report describes two candidate lactate-utilizing bacterial species that were identified using a batch-culture system, then characterized using metagenomics to assess their respective metabolic potential.

## 2. Materials and Methods

### 2.1. Sample Collection and Rumen Culture Experiments

All methods involving animals were in accordance with standard practices that were approved by the South Dakota State University Institutional Animal Care and Use Committee (IACUC; protocol 15-028E). Cannulated beef cows from a research herd at the South Dakota State University Cow-Calf Education and Research Facility were used as rumen fluid donors. These cows were fed the same pasture hay or haylage as the rest of the herd during the time that the study was conducted. Two experiments, each with a different rumen fluid donor, were performed as part of this study, and the same protocol was followed for each experiment. Individual cultures were set up with approximately 2.2 L of fresh rumen fluid contained in laboratory-scale bioreactors (Chemglass Life Sciences, Vineland, NJ, USA), leaving a headspace volume of approximately 0.8 L. Cultures were grown at 39 °C, with continuous agitation (150 rpm), for 14 days. Anaerobic conditions were maintained by a sealed lid, with a single port (4 mm diameter) allowing the release of excess biogas to avoid pressure build-up. Three experimental cultures were supplemented with lactate (20 mL/L, Sodium DL-lactate solution (60%), Catalog# L1375, Sigma, St. Louis, MO, USA), while no substrate was added to the control culture. A volume of approximately 15 mL was collected from the rumen fluid inoculum (D0) as well as from each experimental and control culture on days 7 (D7) and 14 (D14). Samples were stored frozen at −20 °C until analyzed. One of the triplicate lactate-supplemented cultures from Experiment 1 was not analyzed due to fungal contamination.

### 2.2. Microbial Genomic DNA Purification and PCR Amplification of the 16S rRNA Gene

The procedure for the extraction of microbial genomic DNA from 250 µL of rumen or culture samples by the repeated bead beating plus column method was performed as previously described [32]. Briefly, lysis by bead beating in extraction buffer (0.5 M NaCl, 50 mM Tris. HCl, 50Mm EDTA, 4% SDS) was followed by extraction with 10 M ammonium acetate, then by isopropanol precipitation. After its recovery by centrifugation, DNA was purified with the QIAamp DNA Stool Kit (QIAGEN, Hilden, Germany) according to the standard instructions of the manufacturer. Using the 27F [33] and 519R [34] primers, the V1-V3 region of the 16S rRNA gene was amplified by PCR with the Phusion Taq DNA polymerase (ThermoFischer Scientific, Waltham, MA, USA). The amplification protocol, performed on a 2720 Thermo Cycler (Thermo Fischer Scientific), was initiated with a ‘hot start’ (98 °C, 3 min), followed by 35 successive cycles of denaturation (98 °C, 30 s), annealing (50 °C, 30 s), and elongation (72 °C, 30 s), then ending with a final synthesis period (72 °C, 10 min). Agarose gel electrophoresis was used to assess the quality of the PCR-generated DNA (expected length of approximately 500 bp), which was then recovered with the QiaexII Gel extraction kit (Qiagen, Hilden, Germany). Gel-purified amplicons were subsequently sequenced using an Illumina Miseq (2 × 300) platform (Molecular Research DNA, Shallowater, TX, USA).

### 2.3. Bioinformatics Pipeline for 16S rRNA Gene-Based Composition Analysis

Unless specified, the bioinformatics analysis was performed using a set of Perl scripts that were developed in-house (available upon request). The raw sequence reads for the 16S rRNA gene V1-V3 amplicons that were generated by Molecular Research DNA (Shallowater, TX, USA) were assembled into contigs from overlapping paired-end reads generated from the same flow cell clusters. Quality screening was performed using three filtering steps: intact nucleotide sequences for both primers (27F and 519R), a length between 400 and 580 bp, as well as an average Phred quality score of at least Q33. After quality filtering, sequences were aligned, then clustered into operational taxonomic units (OTUs) at a 4% sequence dissimilarity cutoff. It is important to note that 3%, which is the most commonly used clustering threshold, was initially determined based on full-length 16S rRNA gene sequences. Therefore, it may not be suitable for the analysis of shorter regions because sequence dissimilarity varies across the 16S rRNA gene. Considering that 3% is the most commonly used cutoff for the V4 or V4–V5 regions, which are regions that are the least dissimilar, then a higher cutoff should be acceptable for the V1–V3 region because V1 is the most variable region of the 16S rRNA gene [35,36]. Following clustering, OTUs were screened to remove sequence artifacts as previously described [37], which included chimeric sequences [38,39]. Taxonomic assignment of curated OTUs was performed using RDP Classifier (Ribosomal Database Project) [40] and BLAST [41]. Read count yields and OTU tables for both experiments are provided in Appendix A.

### 2.4. Metagenomics Analysis

The rumen fluid cultures with the highest abundance of Bt-01708_Bf and Bt-01899_Ap, respectively, were used as representative samples for metagenomics studies to assemble the genomes of the OTUs of interest. Purified microbial genomic DNA (extracted as described above) was used as a template for ‘shotgun’ sequencing with an Illumina Miseq (2 × 250) platform (Molecular Research DNA, Shallowater, TX, USA). Using a set of in-house-developed Perl scripts, raw datasets were first filtered to keep reads with a length of at least 200 bp, which were then used for contig assembly.

Considering that the samples selected for metagenomics were from complex bacterial communities (as assessed by 16S rRNA gene analysis), it was assumed that each contig set included assemblies that did not belong to the most abundant OTUs of interest. A screening strategy using blastp [41] was then implemented to identify the contigs most likely to belong to the OTUs of interest based on their closest valid relative (Bt-01708_Bf: *Butyrivibrio fibrisolvens*; Bt-01899_Ap: *Anaerococcus prevotii*). Translated protein sequences were generated by RAST (Rapid Annotations using Subsystems Technology) [42] from each contig set. For screening of the Bt-01708_Bf coding sequences, the proteome for *B. fibrisolvens* strain DSM 3071 was used as a reference (Genbank Accession: GCF_900129945.1) because of its NCBI designation as a representative genome for the species. For screening of the Bt-01899_Ap coding sequences, the proteomes from three strains of *A. prevotii* were combined, as there was no recommended NCBI representative genome for this species (Strain NCTC11806, Genbank Accession: GCF_900445285.1; Strain ACS-065-V-Col13, Genbank Accession: GCF_000191725.1; Strain DSM 20548, Genbank Accession: GCF_000024105.1). Taking into consideration the phylogenetic distance between the OTUs of interest and their respective closest match, as estimated by nucleotide sequence identity for the 16S rRNA gene, optimal thresholds for filtering the blastp search results were determined empirically. To this end, various combinations of cutoffs for amino acid sequence identity and alignment coverage were tested. It was determined that thresholds of 50% for amino acid sequence identity and 70% for alignment length were optimal, as they resulted in contig sets with adequate genome coverage based on ribosomal protein representation and combined length (see Section 3). Fasta sequence files with the sets of contig for each OTU are provided as Appendix A (Bt-01708_Bf) and Appendix A (Bt-01899_Ap). Translated protein sequences from Bt-01708_Bf and Bt-01899_Ap contig sets were then annotated using RAST [42], and the predicted enzymes of interest were assigned to metabolic pathways using KEGG pathways as a model reference [43].

### 2.5. Availability and Accession of Next Generation Sequencing Data

Raw sequence data are available from the NCBI Sequence Read Archive under Bioproject PRJNA929221.

## 3. Results

### 3.1. Identification of Candidate Lactate Utilizers from Rumen Fluid

Two experiments using different donors were performed to identify candidate lactate utilizers from the rumen fluid of beef cows. The bacterial composition of the rumen inoculum prior to culturing with lactate was consistent with previously reported studies, as a predominance of Bacteroidetes (Prevotellaceae) and Firmicutes (Lachnospiraceae and Ruminococcaceae) was observed [44,45,46,47,48,49] (Figure 1A). Rumen fluid donors were found to share 617 OTUs, which collectively represented 54.0% of sequence reads from Donor 1 and 31.5% of sequence reads from Donor 2 (Figure 1B). When compared to their respective control culture that was not supplemented with lactate, certain OTUs were found to be in higher abundance in lactate-supplemented rumen fluid cultures. In Experiment 1, OTU Bt-01708_Bf was found to be in the highest abundance in two lactate-supplemented cultures at day 14 (7.6% and 33.7% vs. 0.005%; Appendix A). The closest valid relative to Bt-01708_Bf was *B. fibrisolvens* [50], with a sequence identity of 89.0% based on 16S rRNA gene-based analyses. In Experiment 2, Bt-01899_Ap was found to be the most abundant OTU in lactate-supplemented cultures at day 14, with three cultures each showing a relative abundance that was higher than in the non-supplemented control (0.3%, 2.5% and 46.7% vs. 0.05%; Appendix A). The closest valid relative of Bt-01899_Ap was *A. prevotii* (95.3% sequence identity) [51].

### 3.2. Exploring the Metabolic Potential of OTU Bt-01708_Bf

To gain further insight on the metabolic capabilities of the lactate-enriched OTUs, a metagenomics approach was used. As it was reasonable to assume that sequence datasets generated from samples with a high abundance of an OTU would have a higher abundance of sequence reads from this OTU, higher sequence representation would, in turn, be favorable for the assembly of contigs that would provide sufficient coverage of the OTU’s genome to determine its metabolic potential. In this context, the culture with the highest abundance of Bt-01708_Bf (33.7%) was selected for metagenomics analysis. From a total of 7,006,884 reads generated from this sample, 711 contigs were assembled for a combined total length of 3,436,790 bp; individual contig lengths ranged from 302 bp to 95,784 bp, with an N50 of 2267 bp. Since Bt-01708_Bf was most closely matched to *B. fibrisolvens*, a blastp-based strategy was used to identify contigs that would be most likely to correspond to this OTU. Using this approach, 313 contigs were selected for further analysis; their combined length was 2,717,256 bp, with individual lengths ranging between 407 and 95,784 bp (N50 = 5626 bp) (Appendix A). Considering the phylogenetic distance between Bt-01708_Bf and *B. fibrisolvens,* as assessed by 16S rRNA gene comparisons, it was assumed that predicting the genome size of this OTU based on available genome data from strains of *B. fibrisolvens* would be unreliable. To assess the extent of the coverage of the 313 contig dataset for Bt-01708_Bf, a survey of encoded ribosomal proteins was performed. This analysis revealed that 20 of 21 ribosomal proteins from the small subunit had been found (missing subunit: S2p) and that 28 of 33 ribosomal proteins from the large subunit had been identified (missing subunits: L9p, L20p, L25p, L31p and L35p) (Appendix A). Together, these results indicated that, while incomplete, the 313 contig dataset would offer sufficient coverage to assess the metabolic potential of Bt-01708_Bf.

An analysis of the predicted proteome encoded by the 313 contig dataset was conducted to identify enzymes and proteins that would be involved in metabolic pathways of interest. A total of 1270 coding sequences were annotated to proteins of known functions, while 1365 coding sequences were designated as ‘hypothetical’ proteins. For substrate utilization, lactate and malate were predicted to be metabolized by Bt-01708_Bf. Indeed, the presence of coding sequences for a lactate permease and a malate permease supported that these substrates could be acquired from the extracellular environment (Appendix A). Furthermore, the identification of three coding sequences for distinct lactate dehydrogenases and one coding sequence for a malic enzyme indicated that both compounds could be metabolized into pyruvate (Figure 2). Monosaccharides were also predicted as substrates for Bt-01708_Bf since sequences coding for an endoglucanase (EC 3.2.1.4) and an O-glycosyl hydrolase (GH2 family; EC 3.2.1.-) were identified (Appendix A). Glycoside hydrolase enzymes from these families are involved in the cleavage of (1->4)-beta-D-glycosidic linkages and of glycosidic bonds between carbohydrate and non-carbohydrate moieties, respectively, suggesting the potential to hydrolyze cellulose and to release carbohydrate groups from glycoproteins. However, it should be noted that the ability to cleave (1->4)-beta bonds has been reported to be sufficient for the hydrolysis of soluble cellodextrins in certain bacterial species that are unable to break down cellulose [52]. A glycoside hydrolase from the GH32 family was also identified; as a group, these enzymes are involved in the hydrolysis of a wide range of oligosaccharides. Accordingly, transporters for various types of sugars were found to be encoded in the contig set (Appendix A), as well as most enzymes for glycolysis and glycogen synthesis, indicating that glucose could be metabolized to pyruvate (Figure 2) or stored as glycogen (Appendix A). From pyruvate, the end products formate, acetate and butyrate could be generated by Bt-01708_Bf based on the proteome encoded by the contig set (Figure 2 and Figure 3). The identification of candidate proteases, as well as amino acid transporters, indicated that Bt-01708_Bf would have the necessary enzymes and protein complexes to acquire amino acids from breaking down proteins in the extracellular environment (Appendix A). Quinate, a plant metabolite, was also predicted to be a substrate for Bt-01708_Bf as a complete pathway for metabolizing quinate into chorismate was found (Figure 4); chorismate could, in turn, serve as a precursor for the synthesis of tryptophan. Other potential substrates, as indicated by the presence of ABC-type transport systems, included taurine and polyamines (e.g., spermidine or putrescine) (Appendix A).

Notably, enzymes and proteins involved in generating chemiosmotic gradients, as well as ATP by oxidative phosphorylation, were found to be encoded in the contig subset for Bt-01708_Bf. Indeed, all subunits of a putative NADH ubiquinone oxidoreductase complex (subunits A-N; EC 7.1.1.2) were found, as well as all subunits of a predicted ATP synthase complex (F1: alpha, beta, gamma, delta, epsilon; F0 subunits a, b, c) (Appendix A).

### 3.3. Exploring the Metabolic Potential of OTU Bt-01899_Ap

Using the same metagenomics approach, a total of 7,443,430 reads were generated from the culture with the highest abundance of OTU Bt-01899_Ap (46.7%), which were used to assemble 628 contigs. This contig set had a combined length of 3,383,155 bp, with a range of 315–171,084 bp, and an N50 of 2100 bp. Based on the nucleotide sequence of its 16S rRNA gene, Bt-01899_Ap was predicted to correspond to an uncultured species of the genus *Anaerococcus*. Using the available genomes of 11 species from this genus as comparisons, the genome size of OTU Bt-01899_Ap was predicted to range between 1.8 and 2.27 Mbp. Since the combined lengths of the contigs assembled from this sample were greater than the expected genome length of the OTU, a blastp-based approach was used to identify contigs that would be most likely to correspond to the OTU of interest. From this analysis, a subset of 120 contigs most likely matched to Bt-01899_Ap were selected; these ranged between 562 and 171,084 bp in length (N50 = 7213 bp) and had a combined length of 1,836,139 bp (Appendix A).

A full-length 16S rRNA sequence (1540 bp) was identified in one of the selected contigs (Appendix A). The sequence of OTU Bt-01899_Ap was 99.4% identical to the V1-V3 region of the assembled full-length 16S rRNA gene. The same top match (*A. prevotti*, NR_074575.1) was found using a blastn search against the NCBI 16S rRNA ‘refseq_seq’ database for both Bt-01899_Ap (V1–V3 region, 95.3% sequence identity) and the assembled 16S rRNA gene (95.5%). An assessment of ribosomal proteins revealed that 19 of 21 ribosomal proteins from the small subunit were encoded in the contig subset (missing subunits: S1p and S9p) and that 25 of 33 ribosomal proteins from the large subunit had been found (missing subunits: L7/L12p, L13p, L19P, L21, L25, L27P, L31P and L34P) (Appendix A).

The combined contig lengths, identification of a full-length 16S rRNA sequence, as well as ribosomal protein representation supported that the 1.8 Mbp contig set would be suitable to assess the metabolic potential of Bt-01899_Ap. A total of 871 coding sequences were annotated to proteins of known functions, while 1343 coding sequences were designated as ‘hypothetical’ proteins. Enzymes and other proteins encoded in this contig subset indicated that glucose would likely be metabolized by OTU Bt-01899_Ap. Indeed, three coding sequences for alpha-amylase enzymes, including one with a signal peptide, suggested that OTU Bt-01899_Ap could hydrolyze starch (Appendix A). The identification of subunits for a ‘maltodextrin ABC transporter’ (subunits Mdx E, Mdx F, Mdx G and Mdx K; Appendix A) as well as most glycolytic enzymes (Figure 2) further indicated that glucose in the form of disaccharides or oligosaccharides could be acquired from the extracellular environment to be metabolized to pyruvate. Pyruvate could then, in turn, be used to generate end products such as formate, acetate, lactate or butyrate (Figure 2 and Figure 3). An alternative use for glucose was also predicted to be storage in the form of glycogen (Appendix A). Conversely, the identification of coding sequences for phosphoenolpyruvate synthase, which can convert pyruvate into phosphoenolpyruvate (PEP), as well as for fructose-1,6-bisphosphatase, which converts fructose 1,6 bisphosphate to fructose 6-P (Figure 2), indicated that Bt-01899_Ap could produce glucose 6-P through gluconeogenesis as bi-directional glycolytic enzymes would be capable of performing the other metabolic reactions required for the process. The identification of other PTS systems further suggested that Bt-01899_Ap had the potential to use other monosaccharides as substrates, such as mannose or fructose (Appendix A). Identification of an L-lactate dehydrogenase (EC 1.1.1.27) indicated the ability to either produce or utilize lactate (Figure 2). Contigs were also predicted to include genes for candidate TRAP transporters (Appendix A), a family of membrane-associated proteins that include members that can transport lactate. Coding sequences for proteases were found, along with various amino acid transporters, indicating the ability to acquire amino acids from extracellular sources (Appendix A). All six subunits of a putative Rnf complex were identified, as well as components of a putative V-type ATP synthase (identified subunits A, B, D, E, I, J, K; missing subunits: C, F, G/H), indicating the potential to generate chemiosmotic gradients as well as to generate ATP by oxidative phosphorylation, respectively (Appendix A). The presence of coding sequences for ferredoxin and hydrogenase subunits (HydE, HydF and HydG) was indicative of the capacity to produce hydrogen gas (H_2_) from metabolic hydrogen (H) (Appendix A).

## 4. Discussion

The accumulation of lactate as a result of feeding diets with a high inclusion of easily fermentable feedstuffs is one of the main factors that can exacerbate the development of acidosis, a commonly occurring digestive disorder in intensive ruminant production systems [10]. While rumen microbial communities include lactate-utilizing symbionts, their ability to mitigate the accumulation of lactate has limitations. Consequently, a common strategy to prevent the onset of acidosis is to gradually adapt animals to higher inclusions of concentrates in their diet. This way, populations of lactate-utilizing microorganisms can proliferate to reach an abundance level that is sufficient to keep lactate levels in check. While implemented to ensure animal health and welfare, this management practice is not ideal from an economics perspective as it incurs costs from delays in reaching optimal animal performance levels.

An improvement on this strategy would be to modulate populations of lactate utilizers in order to increase their capacity to metabolize lactate, which could be achieved through manipulating their growth and/or activity. Most efforts to gain a better understanding of how lactate utilizers can counter the development of acidosis have focused on *M. elsdenii*, formerly known as *Peptostreptococcus elsdenii* or ‘microorganism LC’. As this gut symbiont was one of the earliest identified, it has consequentially been one of the most studied lactate utilizers [53,54,55,56]. *M. elsdenii* can metabolize lactate into propionate through the acrylate pathway [57], and while commercial strains have been developed for use in the livestock industry, their efficacy remains to be improved and optimized. As an estimated 95% of ruminal bacterial species have yet to be assigned a function [30], it is likely that other lactate-metabolizing symbionts remain to be characterized in this complex microbial ecosystem. Accordingly, we were able to identify previously unknown ruminal bacteria that exhibited characteristics of lactate utilizers. Indeed, both OTUs were found in higher abundance in rumen fluid cultures that were supplemented with lactate as the only added substrate, and contigs generated for each OTU included genes encoding lactate dehydrogenase enzymes, which would catalyze the conversion of lactate into pyruvate [28,58]. Based on predictions from gene annotation, the OTUs could acquire lactate from the extracellular environment through the function of lactate permeases (Bt-01708_Bf) or TRAP-type transporters (Bt-01899_Ap). In both OTUs, potential end products from metabolizing lactate could be formate, acetate or butyrate. One of the main differences in their metabolic capacity in comparison with *M. elsdenii* and *S. ruminantium* subsp. *lactilytica* would be their inability to produce propionate, whether through the acrylate [57] or succinate pathway [59], respectively.

Considering the low relative abundance of Bt-01708_Bf and Bt-01899_Ap in the rumen fluid used as inoculum in this study, it is possible that these bacterial OTUs were only transient residents in the rumen. Their presence could have been the result of being ingested from feed such as haylage, which supports their own bacterial communities. To our knowledge, members of the genera *Butyrivibrio* and *Anaerococcus*, to which Bt-01708_Bf and Bt-01899_Ap are affiliated, have not been reported as members of bacterial communities in silage [60,61], but they are commonly found in gut environments. Using a culturing approach, Ghali et al. (2011) [62] have previously reported the characterization of lactate-utilizing isolates from the foregut of feral camels that were related to *B. fibrisolvens*. Thus, despite the substantial phylogenetic distance between *B. fibrisolvens* and OTU Bt-01708_Bf, as estimated by 16S rRNA gene sequence comparisons, both appear to share the ability to utilize lactate. Notably, certain strains of *B. fibrisolvens* are capable of elevated lactate production from metabolizing glucose [63], indicating that this bacterial species is both a primary and secondary fermenter; whether OTU Bt-01708 possesses a similar capacity remains to be determined. In contrast, lactate utilization from *Anaerococcus* species does not appear to have previously been reported, as assessed by a database search of available scientific literature at the time of submission of this report. Members of the genus *Anaerococcus* have been isolated from a number of different habitats, such as skin, the oral cavity, and the gut [64]. They have been described as able to use various monosaccharides as substrates for fermentation [65,66,67], which is consistent with the metagenomics analysis presented in this report.

The OTUs characterized in this study shared the ability to grow in culture with lactate provided as the only substrate, an observation that was supported by the identification of genomic sequences encoding enzymes and transporters that would be required to perform these functions. As coding sequences for lactate racemase or d-lactate dehydrogenase were not found in the contigs analyzed, it remains to be determined whether the OTUs identified in this study would be able to metabolize both l- and d-lactate isomers. In addition, both OTUs were predicted to generate formate, acetate and butyrate as end products, as well as have the ability to store glucose intracellularly as glycogen. However, metagenomics analyses revealed major differences in metabolic potential between them, which suggested that they may occupy distinct niches in the rumen. For instance, the contigs assigned to Bt-01899_Ap encoded for three alpha-amylase enzymes that hydrolyze (1->4)-alpha-D-glycosidic bonds, indicating that this OTU likely hydrolyzes starch from feedstuffs as its main source of substrate (glucose) to ferment. In contrast, alpha-amylase enzymes were not identified in the contig set assigned to OTU Bt-01708_Bf, which instead had coding sequences for enzymes that could hydrolyze (1->4)-beta-d-glycosidic bonds from compounds such as polysaccharides or other types of macromolecules. However, as mentioned above, the identification of a potential glycoside hydrolase that could cleave (1->4)-beta bonds is not sufficient evidence of the capacity to metabolize polysaccharides, as certain bacterial species capable of hydrolyzing cellodextrins are unable to break down cellulose [52]. Bt-01708_Bf appeared to have more metabolic flexibility than Bt-01899_Ap, with pathways for the utilization of other types of substrates, such as malate, quinate, taurine and polyamines.

Generating ATP is an essential function for all organisms, and this can be accomplished through substrate-level phosphorylation (e.g., pyruvate kinase) or oxidative phosphorylation (ATP synthase). Both OTUs were found to encode the necessary enzymes and proteins to use these mechanisms. Contigs for Bt-01708_Bf encoded all individual subunits (NuoA to NuoN) of an NADH:ubiquinone oxidoreductase, also known as bacterial respiratory complex I. These types of complexes can convert the energy stored in reducing equivalents (e.g., NADH) into a ‘proton motive force’, i.e., an ion gradient across the cellular membrane [68]. The energy from the proton motive force can then be converted to ATP by ATP synthase [68]. In the context of lactate utilization, the conversion of lactate to pyruvate by lactate dehydrogenase would generate NADH, which is the main entry point for electrons in these types of bacterial respiratory complexes [68]. Contigs for Bt-01899_Ap were found to encode all subunits of a putative Rnf complex, which can convert the energy from NADH into transporting Na^+^ across the plasma membrane, thereby generating a chemiosmotic gradient [69,70]. In Bt-01899_Ap, the conversion of reducing equivalents into H_2_ by a ferredoxin–hydrogenase complex could also be used as an alternative mechanism for the conversion of NADH [71].

## 5. Conclusions

In this report, we have provided evidence of the identification of previously uncharacterized ruminal bacterial species that can metabolize lactate. This included a combination of a biological assay (lactate-supplemented cultures), microbial composition analysis (16S rRNA gene profiling) and functional annotation (metagenomics). While neither OTU was predicted to metabolize lactate into propionate, a signature metabolic activity of *M. elsdenii*, both were found to encode the necessary enzymes to produce butyrate, an SCFA that is also beneficial for the host. Based on its metabolic capabilities, Bt-01708_Bf may show more promise as a lactate utilizer to target for controlling lactate levels than Bt-01899_Ap as it appeared to lack typical enzymes (e.g., cellulases) to hydrolyze structural polysaccharides, which would be consistent with a microorganism dependent on lactate as a substrate. This candidate bacterial species also showed greater potential flexibility in substrate utilization than Bt-01899_Ap. However, future research will be required to determine whether either candidate bacterial species could be targeted or developed for alleviating acidosis. A continued effort towards the characterization of lactate utilizers will help in establishing distinct categories or subgroups of ruminal bacterial species that can metabolize lactate based on other metabolic capabilities. This will ultimately contribute to a better understanding of metabolic functions and events that either promote or help mitigate the onset of SARA and the incidence of acidosis.

## Figures and Tables

**Figure 1 microorganisms-11-00658-f001:**
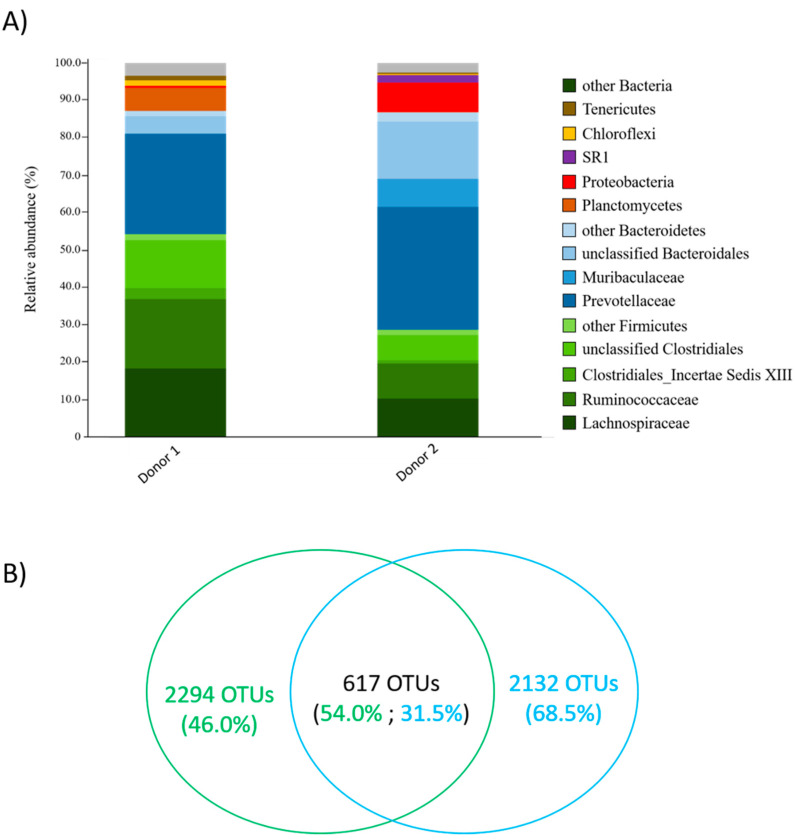
Comparison of the bacterial communities between rumen fluid donors used in this study. (**A**) Taxonomic composition shows representation of the main phyla (different colors) and families (shades of the same color) for Firmicutes (green) and Bacteroidetes (blue). (**B**) Venn diagram showing the number of OTUs that were either shared by both rumen fluid donors or unique to each donor. Percentages represent the relative abundance of sequence reads for each category of OTU (shared or unique), with Donor 1 shown in green and Donor 2 shown in blue.

**Figure 2 microorganisms-11-00658-f002:**
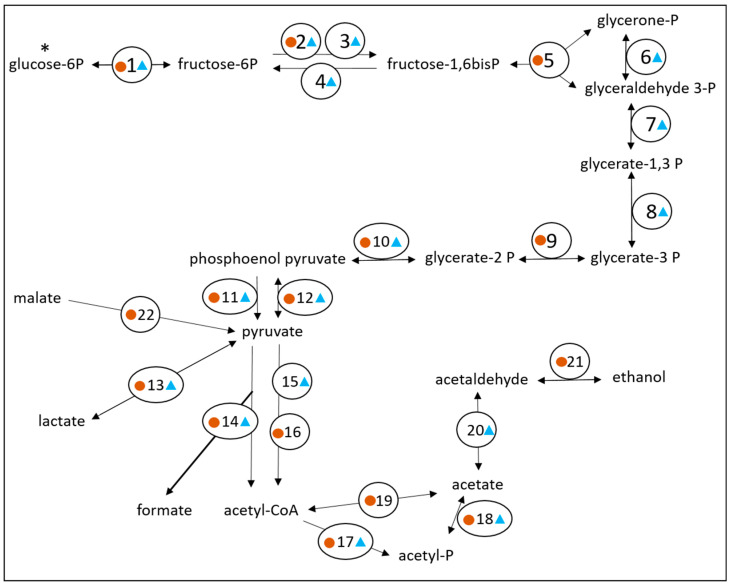
Proposed glycolytic pathway for Bt-01708_Bf and Bt-01899_Ap. Coding sequences for enzymes identified in the Bt-01708_Bf contig set are labeled with a brown circle, while coding sequences for enzymes identified in the Bt-01899_Ap contig set are labeled with a blue triangle. As the metagenomics analysis revealed PTS systems, while hexokinases were not found (Appendix A), the pathway is shown as starting with glucose-6P (*). Key for enzyme legend: (1) glucose-6-phosphate isomerase (EC 5.3.1.9); (2) 6-phosphofructokinase (EC 2.7.1.11); (3) pyrophosphate-dependent fructose 6-phosphate-1-kinase (EC 2.7.1.90); (4) fructose-1,6-bisphosphatase (EC 3.1.3.11); (5) aldolase (EC 4.1.2.13); (6) triosephosphate isomerase (EC 5.3.1.1); (7) glyceraldehyde-3-phosphate dehydrogenase (EC 1.2.1.12); (8) phosphoglycerate kinase (EC 2.7.2.3); (9) phosphoglycerate mutase (EC 5.4.2.11); (10) enolase (EC 4.2.1.11); (11) pyruvate kinase (EC 2.7.1.40); (12) phosphoenolpyruvate synthase (EC 2.7.9.2); (13) L-lactate dehydrogenase (EC 1.1.1.27); (14) pyruvate formate lyase (EC 2.3.1.54); (15) pyruvate-flavodoxin oxidoreductase (EC 1.2.7.-); (16) pyruvate dehydrogenase (EC 1.2.4.1); (17) phosphate acyltransferase; (18) acetate kinase (EC 2.7.2.1); (19) acetyl-CoA synthatase (AMP forming) (EC 6.2.1.1); (20) aldehyde dehydrogenase (EC 1.2.1.3); (21) alcohol dehydrogenase (EC 1.1.1.1); (22) NADP-dependent malic enzyme (EC 1.1.1.40).

**Figure 3 microorganisms-11-00658-f003:**
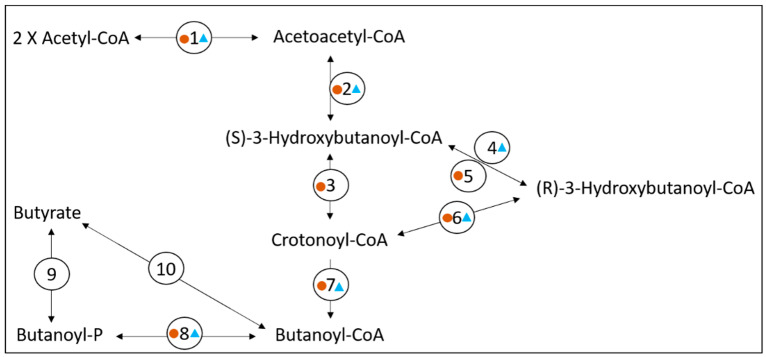
Proposed butyrate synthesis pathway for Bt-01708_Bf and Bt-01899_Ap. Coding sequences for enzymes identified in the Bt-01708_Bf contig set are labeled with a brown circle, while coding sequences for enzymes identified in the Bt-01899_Ap contig set are labeled with a blue triangle. Key for enzyme legend: (1) acetyl-CoA C-acetyltransferase (EC 2.3.1.9); (2) 3-hydroxybutyryl-CoA dehydrogenase (EC 1.1.1.157); (3) enoyl-CoA hydratase (EC 4.2.1.17); (4) 3-hydroxyacyl-CoA dehydrogenase (EC 1.1.1.35); (5) acetoacetyl-CoA reductase (EC 1.1.1.36); (6) 3-hydroxybutyryl-CoA dehydratase (EC 4.2.1.55); (7) butyryl-CoA dehydrogenase-electron-transferring flavoprotein complex (Bcd-EtfAB); (8) phosphate acyltransferase; (9) butyrate kinase (EC 2.7.2.7); (10) acetate CoA/acetoacetate CoA-transferase alpha/beta subunits (EC:2.8.3.8/2.8.3.9).

**Figure 4 microorganisms-11-00658-f004:**
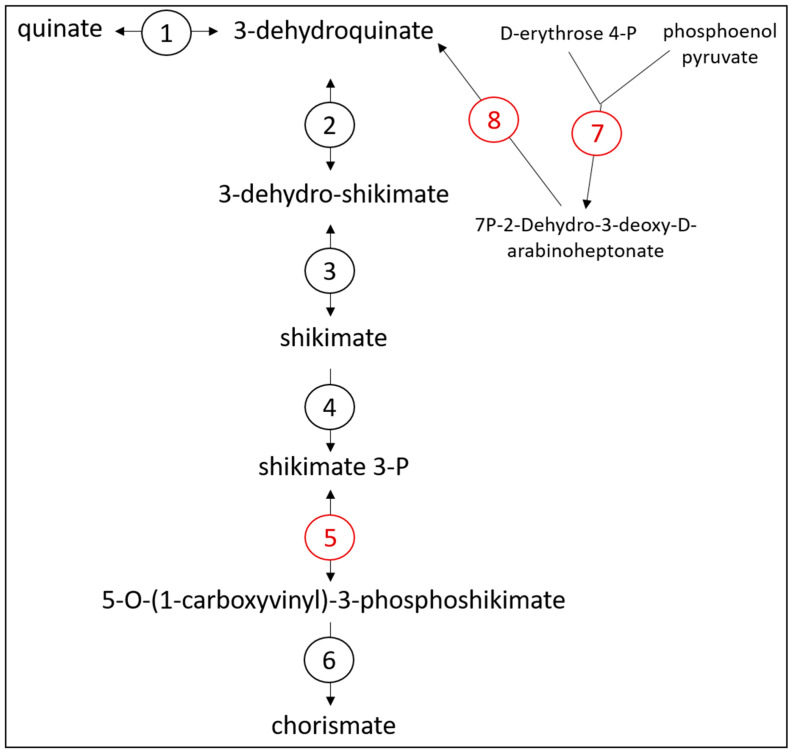
Proposed chorismate synthesis pathway for Bt-01708_Bf. The main precursors for chorismate synthesis are quinate (plant compound) or phosphoenol pyruvate (glycolysis) + D-erythrose 4-P (pentose phosphate pathway). Enzymes labeled in black were encoded in the contig set for Bt-01708_Bf, while enzymes shown in red were not found. Key for enzyme legend: (1) shikimate/quinate 5-dehydrogenase (EC 1.1.1.282); (2) 3-dehydroquinate dehydratase II (EC 4.2.1.10); (3) shikimate 5-dehydrogenase I alpha (EC 1.1.1.25); (4) shikimate kinase I (EC 2.7.1.71); (5) 5-enol-pyruvylshikimate-3-phosphate synthase (EC 2.5.1.19); (6) chorismate synthase (EC 4.2.3.5); (7) 3-deoxy-7-phosphoheptulonate synthase (EC 2.5.1.54; (8) 3-dehydroquinate synthase (EC 4.2.3.4).

## Data Availability

Raw sequence data are available from the NCBI Sequence Read Archive under Bioproject PRJNA929221 (Home-SRA-NCBI (nih.gov)).

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
