# Peer review of "Metagenomics-Based Analysis of Candidate Lactate Utilizers from the Rumen of Beef Cattle"

_microorganisms, 2023, doi:10.3390/microorganisms11030658_

Round 1

Reviewer 1 Report

The authors used 16S rRNA gene-based identification of two bacterial OTUs, Bt-01708_Bf (89.0% identical to Butyrivibrio fibrisolvens) and Bt-01899_Ap (95.3% identical to Anaerococcus prevotii), and analyses the predicted proteomes from metagenomics-assembled contigs assigned to these candidates ruminal bacterial species, and revealed genes encoding lactate dehydrogenase, a putative lactate transporter, as well as pathways for the production of short-chain fatty acids (formate, acetate and butyrate), and the synthesis of glycogen. The present study would contribute to the continued characterization of ruminal bacterial species which can metabolize lactate.

1. How many cannulated beef cows were used as rumen fluid donors?

2. I am worried that the author only used two lactate-supplemented cultures from experiment 1 for analyses. Is the result generally representative?

3. I don't understand how the author's research on these two strains of lactic acid utilization bacteria can help alleviate subacute acidosis. The author needs to further analyze how these two bacteria can be used to alleviate subacute acidosis.

Author Response

Response to comments from Reviewer 1

The authors used 16S rRNA gene-based identification of two bacterial OTUs, Bt-01708_Bf (89.0% identical to Butyrivibrio fibrisolvens) and Bt-01899_Ap (95.3% identical to Anaerococcus prevotii), and analyses the predicted proteomes from metagenomics-assembled contigs assigned to these candidates ruminal bacterial species, and revealed genes encoding lactate dehydrogenase, a putative lactate transporter, as well as pathways for the production of short-chain fatty acids (formate, acetate and butyrate), and the synthesis of glycogen. The present study would contribute to the continued characterization of ruminal bacterial species which can metabolize lactate.

Thank you very much for your review and positive assessment of our manuscript.

R1-C1.

  1. How many cannulated beef cows were used as rumen fluid donors?

A total of two cannulated cows were used in the study, one for each batch culture experiment performed (see lines 89-90).

R1-C2.

  1. I am worried that the author only used two lactate-supplemented cultures from experiment 1 for analyses. Is the result generally representative?

As a standard procedure, we typically use three replicate cultures for each experimental condition tested (‘conditions tested’ could be different substrates or different concentrations of a test compound). Triplicate cultures ensure that if one becomes contaminated or does not respond to the treatment, the remaining two cultures can allow us to determine whether there is a response or not. While it is not ideal, we move forward with our analysis if at least two replicates support a response to the treatment, as was the case for experiment 1 (both lactate-supplemented cultures showed enrichment compared to the non-supplemented control; lines 193-195 in the original MS, lines 194-195 in the revised MS). Since our approach consists of assessing the metabolic potential of enriched bacteria by building partial genomes, we can confirm that the enrichment was representative by matching the metabolic activity observed in culture (e.g. proliferation in the presence of a substrate) with the identification of genes encoding for enzymes or transporters that would be required for metabolizing the test substrate.

R1-C3.

  1. I don't understand how the author's research on these two strains of lactic acid utilization bacteria can help alleviate subacute acidosis. The author needs to further analyze how these two bacteria can be used to alleviate subacute acidosis.

We completely agree with the reviewer that the candidate bacterial species identified in this study would need to be further analyzed and developed in order to be used to mitigate acidosis. It was not our intent to imply that the results presented in this manuscript would be sufficient to be used directly for mitigating acidosis. Rather, the aim of the study was to identify uncharacterized lactate utilizers in order to gain more insight and information on lactate utilization in the rumen; discussions on mitigating acidosis are meant to provide a broader context to the research that was performed.

While we discuss in the ‘Conclusion’ paragraph the possibility that these OTUs could be used or targeted for mitigating acidosis, our intent was to present this potential as hypothetical. However, to make sure that this is conveyed clearly to the reader, we have added the following sentence to the conclusion paragraph:

(Revised manuscript, lines 485-487) “However, future research will be required to determine whether either candidate bacterial species could be targeted or developed for alleviating acidosis.”

Reviewer 2 Report

The manuscript describes a metagenomic analysis of two candidate lactate-utilizing bacterial species from ruminal enrichment cultures. The authors performed extensive genomic characterization that verifies the presence of genes encoding for proteins critical for lactate utilization, along with other pathways that suggest substantial catabolic versatility of both candidate strains. Overall, this is a well-done, and well-articulated study, that could broaden our concept of ruminal lactate utilization.

            That said, the reviewer does have one major concern: Are these two candidates actual functioning members of the ruminal community? It is quite easy to isolate from the rumen not only normal residents, but also transients that are merely passing through. E. coli is a good example. Another is Clostridium kluyveri. The latter is easily enriched and isolated from the rumen, but has been shown by qPCR  to be present in only very low abundance, and in fact at only one-tenth of its abundance in silages (the primary component of the cow’s feed; see Weimer and Stevenson, doi: 10.1007/s00253-011-3751-z). Inspection of Supplementary Table 1 of the current submission indicates that the authors’ candidate SD_Bt-01708_Bf is present in the inoculum at an abundance of only “4.91E05”; this would seem to be 0.00491%, correct? The abundance of the other candidate is not apparent because (unlike SD_Bt-01708_Bf) it is not flagged in Column I of the Table. Of course, estimating abundances based on 16S gene copy number is complicated by variation in this copy number among species (and, oddly, the authors do not state 16S gene copy number of their two candidates). Nevertheless, the abundance of these candidates seems very low. Because the authors used ruminal inocula from haylage fed animals (rather than from grain-fed animals that would have been more likely to be acidotic and lactate rich), it’s hard to predict how abundant the candidates would have been in an acidotic rumen. Regardless, the authors make no comments on this low abundance, nor do they consider the fact that the ultimate source and primary habitat of their candidates may well have been the lactate-rich haylage fed to the cows (perhaps like Lactobacillus, which is very abundant in silage but not particularly abundant, and probably not particularly active, in the rumen itself). On the whole, the lack of consideration of relative abundance, ultimate source, and residence versus transience, is a serious oversight the authors need to address.

Specific comments:

L67: As only some strains of S. ruminantium can ferment lactate, change to “Selenomonas ruminantium subsp. lactilytica”.

L88-89: Given that lactate utilizers are more abundant in the rumens of acidotic cows, it is a little surprising that the authors used inocula from cows that were fed a haylage diet. As haylage would be an ideal habitat for a lactate-utilizing bacterium, it would have been of interest to determine the abundance of the two candidates in the feed of the cows used rumen fluid donors.

L97: Indicate here the concentration of the Na lactate solution (60%, per Sigma catalog).

L245-248: Ability to hydrolyze b-1,4-glycosidic linkages is necessary, but not sufficient, for hydrolyzing cellulose: Many bacteria can hydrolyze these linkages to degrade soluble cellodextrins, but most of these are not cellulolytic (i.e. cannot hydrolyze cellulose). See Russell, doi:10.1128/aem.49.3.572-576.1985

L258: Suggest dropping “abundant”. Quinate is hardly abundant, except in tree bark and coffee beans, neither of which are in typical ruminant diets.

L396: The authors might mention here that the efficacy of these strains is questionable.

L408-409: The authors might add parenthetically that this trait also distinguishes them from S. ruminantium subsp. lactilytica, which produces propionate via the randomizing (succinate) pathway (see Ricke et al., doi:10.3109/10408419609106455).

L410-414: The authors might mention here that B. fibrisolvens is well-known to produce substantial amounts of lactate during carbohydrate degradation (see Diez-Gonzalez et al. doi: 10.1007/s002030050717). Could OTU Bt-01708_Bf  possess the twin property of lactate production and lactate consumption, which would make the organism both a primary and secondary fermenter?

L421-424: Regarding lactate utilization, does either strain have genes encoding either a lactate racemase, or a D-lactate dehydrogenase, that would permit the utilization of both lactate isomers?

L433: Add here cellodextrins, as per comment to L245-248 above.

Minor edits:

L31: Insert “human-“ ahead of “inedible”.

L45, L219: Change “Since” to “Because”.

L152: Insert “from” ahead of “complex”.

Author Response

Response to comments from Reviewer 2

The manuscript describes a metagenomic analysis of two candidate lactate-utilizing bacterial species from ruminal enrichment cultures. The authors performed extensive genomic characterization that verifies the presence of genes encoding for proteins critical for lactate utilization, along with other pathways that suggest substantial catabolic versatility of both candidate strains. Overall, this is a well-done, and well-articulated study, that could broaden our concept of ruminal lactate utilization.

Thank you for your positive assessment, and we truly appreciate the effort that you committed to this review. The recommendations that you have made will greatly help in improving the manuscript. We also appreciate how you have clearly expressed your concerns, which helps us greatly in addressing them.

As you will see, we took the liberty of labelling your comments to facilitate the process of verifying the revisions in the manuscript.

 R2-C1.  That said, the reviewer does have one major concern: Are these two candidates actual functioning members of the ruminal community? It is quite easy to isolate from the rumen not only normal residents, but also transients that are merely passing through. E. coli is a good example. Another is Clostridium kluyveri. The latter is easily enriched and isolated from the rumen, but has been shown by qPCR  to be present in only very low abundance, and in fact at only one-tenth of its abundance in silages (the primary component of the cow’s feed; see Weimer and Stevenson, doi: 10.1007/s00253-011-3751-z). Inspection of Supplementary Table 1 of the current submission indicates that the authors’ candidate SD_Bt-01708_Bf is present in the inoculum at an abundance of only “4.91E05”; this would seem to be 0.00491%, correct? The abundance of the other candidate is not apparent because (unlike SD_Bt-01708_Bf) it is not flagged in Column I of the Table. Of course, estimating abundances based on 16S gene copy number is complicated by variation in this copy number among species (and, oddly, the authors do not state 16S gene copy number of their two candidates). Nevertheless, the abundance of these candidates seems very low. Because the authors used ruminal inocula from haylage fed animals (rather than from grain-fed animals that would have been more likely to be acidotic and lactate rich), it’s hard to predict how abundant the candidates would have been in an acidotic rumen. Regardless, the authors make no comments on this low abundance, nor do they consider the fact that the ultimate source and primary habitat of their candidates may well have been the lactate-rich haylage fed to the cows (perhaps like Lactobacillus, which is very abundant in silage but not particularly abundant, and probably not particularly active, in the rumen itself). On the whole, the lack of consideration of relative abundance, ultimate source, and residence versus transience, is a serious oversight the authors need to address.

We have included a passage in the discussion describing the possibility that these OTUs are not normal residents in the rumen (and we provide further justifications for this addition below):

(Revised manuscript, lines 414-420) “Considering the low relative abundance of Bt-01708_Bf and Bt-01899_Ap in the rumen fluid used as inoculum in this study, it is possible that these bacterial OTUs were only transient residents in the rumen. Their presence could have been the result of being ingested from feed such as haylage, which support their own bacterial communities. To our knowledge, members of the genera Butyrivibrio and Anaerococcus, to which Bt-01708_Bf and Bt-01899_Ap are affiliated, have not been reported as members of bacterial communities in silage [60, 61], but they are commonly found in gut environments.“

Added references:

  1. Yang, F., Zhao, S., Wang, Y., Fan, X., Wang, Y., Feng, C. 2021. Assessment of bacterial community composition and dynamics in alfalfa silages with and without Lactobacillus plantarum inoculation using absolute quantification 16S rRNA Sequencing. Front. Microbiol. 11:629894. doi: 10.3389/fmicb.2020.629894.
  2. Zhang, G., Fang, X., Feng,G., Li, Y., Zhang, Y. 2020. Silage fermentation, bacterial community, and aerobic stability of total mixed ration containing wet corn gluten feed and corn stover prepared with different additives. Animals 10(10), 1775. https://doi.org/10.3390/ani10101775.

One of the risks of a culture-based approach is certainly the enrichment of false positives, which could be contaminants or opportunists. It is then indeed possible that the OTUs identified in this study may not be normal residents of the rumen, but rather transient bacterial species. This would require  that they were

  • ingested in sufficient quantities from forage or haylage to provide a ‘critical mass’ to sustain survival and or proliferation
  • able to maintain their numbers in the rumen in sufficient amounts to be present in the inoculum
  • capable of outgrowing other rumen microbial species in rumen fluid batch cultures supplemented with lactate

There are a few reasons why we are confident that the OTUs described in the manuscript represent legitimate (albeit less abundant) residents of the rumen.

  • The reviewer is correct that the abundance of SD_Bt-01708_Bf in the inoculum was 0.00491% (note that the abundance of the other OTU in its inoculum was 0.5%, see line 200 in either the original or revised manuscripts). Since bacterial cell densities in the rumen are estimated in the range of 1011 / ml to 1013/ ml, the density of OTU SD_Bt-01708 would then be estimated at between 4.9 X 106 and 4.9 X 108 / ml. The highest bacterial cell counts in silage that we have found in available literature are in the order of 107 cfu / g of fresh matter (data from Zhang et al, 2020; doi: 10.3390/ani10101775). In their bacterial composition analysis of silage, Zhang et al also identified 10 prominent genera, but these did not include either Butyrivibrio nor Anaerococcus. We interpret these results as indicating that if the OTUs were present at all in silage samples, their abundance would be very low.
  • The candidate lactate utilizers identified in our study are affiliated to bacterial lineages that have been previously reported in animal-associated microbiomes (e.g. gut or reproductive tract).
  • As our method made use of a batch culture approach, other factors besides the ability to use lactate would act as selection factors, such as competition or cooperation with other ruminal bacteria as well as the ability to thrive under chemical conditions that are very different from the environment provided in silage.

The reviewer also raised concerns regarding the impact of 16S rRNA gene copy number and adjustment for relative abundance. While we were very successful overall in assembling contigs, only one full length 16S rRNA gene was found in the sets of contigs that we analyzed. Assembly of shotgun reads for 16S rRNA genes remains a challenge despite our best efforts, and we agree that it would provide valuable information such as gene copy numbers.

While we agree that differences in copy number affect the accuracy of estimates of bacterial composition, we are not convinced that correction based on phylogenetic affiliations should be applied without reliable genomic information on individual species. Since it is unclear whether 16S rRNA copy numbers are conserved within bacterial lineages, and that most ruminal bacterial species are assigned as ‘uncultured’ species, estimating copy numbers based on genome data from closest valid taxa would run the risk of introducing other biases for copy number. We recommend the work from Starck et al (2021) [https://doi.org/10.1007/s00248-020-01586-7] and from Louca et al (2018) [https://doi.org/10.1186/s40168-018-0420-9], who have concluded that correcting for 16S rRNA copy number ‘remains an unsolved problem’.

Our strategy to minimize the possible impact of variation in 16S rRNA copy number on our interpretation of bacterial composition results is to limit comparisons to within individual OTUs (such as comparing the abundance of an individual OTU across different treatments, for example), and to avoid comparison between different OTUs. If the abundance of an OTU is over- or underestimated because of copy number, then its representation across different samples should correct for itself because the same OTU would have the same number of copies in the different samples.

Specific comments:

R2-C2. L67: As only some strains of S. ruminantium can ferment lactate, change to “Selenomonas ruminantium subsp. lactilytica”.

Thank you for bringing this to our attention. The text now reads (added text underlined):

(Revised manuscript, line 67) “(… ) such as Megasphaera elsdenii and Selenomonas ruminantium subsp. lactilytica, have the ability to metabolize lactate to counter this effect [7, 14, 24-27].”

R2-C3. L88-89: Given that lactate utilizers are more abundant in the rumens of acidotic cows, it is a little surprising that the authors used inocula from cows that were fed a haylage diet. As haylage would be an ideal habitat for a lactate-utilizing bacterium, it would have been of interest to determine the abundance of the two candidates in the feed of the cows used rumen fluid donors.

Considering that our general goal was to identify novel lactate utilizers from the rumen environment, we have two main concerns with using rumen fluid from acidotic cows as inoculum. First, this would likely be replicating a high number of previous research studies on ruminal lactate utilization, thereby increasing the likelihood of identifying previously characterized lactate utilizers. Secondly, the batch culturing approach that we use is effective when the inoculum has a low abundance of a microbial species or functional group of interest, because it then allows the growth and convincing increase in abundance of uncharacterized microbial species in response to a substrate of interest. Therefore, we purposely used rumen fluid from pasture or haylage-fed cows as inoculum by design, because it would contain a lower abundance of lactate utilizers.

R2-C4. L97: Indicate here the concentration of the Na lactate solution (60%, per Sigma catalog).

Thank you for bringing this to our attention. The text now reads (added text underlined):

(Revised manuscript, line 97) “(…) lactate (20 ml/L, Sodium DL-lactate solution (60%), Catalog# L1375, Sigma), while (…)”

R2-C5. L245-248: Ability to hydrolyze b-1,4-glycosidic linkages is necessary, but not sufficient, for hydrolyzing cellulose: Many bacteria can hydrolyze these linkages to degrade soluble cellodextrins, but most of these are not cellulolytic (i.e. cannot hydrolyze cellulose). See Russell, doi:10.1128/aem.49.3.572-576.1985

Thank you for bringing this to our attention. The following sentence has been added:

(Revised manuscript, lines 248-251) “However, it should be noted that the ability to cleave (1->4)-beta bonds has been reported to be sufficient for hydrolysis of soluble cellodextrins in certain bacterial species that are unable to breakdown cellulose [52].”

Reference added:

Russell, J.B. 1985. Fermentation of cellodextrins by cellulolytic and noncellulolytic rumen bacteria. Appl. Env. Microbiol. 49, 572-576.

R2-C6. L258: Suggest dropping “abundant”. Quinate is hardly abundant, except in tree bark and coffee beans, neither of which are in typical ruminant diets.

Agreed. The text now reads:

(Revised manuscript, line 261) “Quinate, a plant metabolite, was also predicted to be a substrate for Bt-01708_Bf, (…)”

R2-C7. L396: The authors might mention here that the efficacy of these strains is questionable.

Agreed. The text now reads (added text underlined):

(Revised manuscript, lines 398-399) “M. elsdenii can metabolize lactate into propionate through the acrylate pathway [56], and, while commercial strains have been developed for use in the livestock industry, their efficacy remains to be improved and optimized.”

R2-C8. L408-409: The authors might add parenthetically that this trait also distinguishes them from S. ruminantium subsp. lactilytica, which produces propionate via the randomizing (succinate) pathway (see Ricke et al., doi:10.3109/10408419609106455).

Thank you for this recommendation. We have revised this passage which now reads:

(Revised manuscript, lines 410-413) “One of the main differences in their metabolic capacity in comparison with M. elsdenii and S. ruminantium subsp. lactilytica would be their inability to produce propionate, whether through the acrylate [57] or succinate pathway [59], respectively.

Reference added:

Ricke, S.C., Martin, S.A. and Nisbet, D.J. 1996. Ecology, metabolism, and genetics of ruminal Selenomonads. Crit. Rev. Microbiol. 22, 27-65.

R2-C9. L410-414: The authors might mention here that B. fibrisolvens is well-known to produce substantial amounts of lactate during carbohydrate degradation (see Diez-Gonzalez et al. doi: 10.1007/s002030050717). Could OTU Bt-01708_Bf  possess the twin property of lactate production and lactate consumption, which would make the organism both a primary and secondary fermenter?

We agree that this information is important to include in the manuscript. Following your recommendation, we have added the following sentence in this section:

(Revised manuscript, lines 425-428) “Notably, certain strains of B. fibrisolvens are capable of elevated lactate production from metabolizing glucose [63], indicating that this bacterial species is both a primary and secondary fermenter; whether OTU Bt-01708 possesses a similar capacity remains to be determined.”

Reference added:

Diez-Gonzalez, F., Bond, D.R., Jennings, E. and Russell, J.B. 1999. Alternative schemes of butyrate production in Butyrivibrio fibrisolvens and their relationship to acetate utilization, lactate production, and phylogeny. Arch. Microbiol. 171, 324-330.

R2-C10. L421-424: Regarding lactate utilization, does either strain have genes encoding either a lactate racemase, or a D-lactate dehydrogenase, that would permit the utilization of both lactate isomers?

We did not find lactate racemase, or a D-lactate dehydrogenase, so it is unclear whether these enzymes were not encoded in the genomes of the OTUs or whether we were unable to build contigs for the chromosomal regions that would encode these enzymes.

Regardless, we agree that this information is important to include in the MS. This sentence was added to this section:

(Revised manuscript, lines 438-440) “As coding sequences for lactate racemase or D-lactate dehydrogenase were not found in the contigs analyzed, it remains to be determined whether the OTUs identified in this study would be able to metabolize both L- and D-lactate isomers.”

R2-C11. L433: Add here cellodextrins, as per comment to L245-248 above.

Agreed. We have added a sentence in this section to incorporate this information:

(Revised manuscript, lines 450-454) “However, as mentioned above, the identification of a potential glycoside hydrolase that could cleave (1->4)-beta bonds is not sufficient evidence of the capacity to metabolize polysaccharides, as certain bacterial species capable of hydrolyzing cellodextrins are unable to breakdown cellulose [52].”

Reference added:

Russell, J.B. 1895. Fermentation of cellodextrins by cellulolytic and noncellulolytic rumen bacteria. Appl. Env. Microbiol. 49, 572-576.

Minor edits:

R2-C12. L31: Insert “human-“ ahead of “inedible”.

Agreed. The text now reads (added text underlined):

(Revised manuscript, line 31) “(…) ability to transform human-inedible plant biomass into products (…)”

R2-C13. L45, L219: Change “Since” to “Because”.

We respectfully disagree. As a matter of style (‘since’ is more formal than ‘because’), we prefer to begin these two sentences with ‘Since’ rather than ‘Because’. The use of ‘Since’ in these sentences also puts more emphasis on the ‘result’ than on the ‘reason’.

R2-C14. L152: Insert “from” ahead of “complex”

Agreed. The text now reads (added text underlined):

(Revised manuscript, line 152) “(…) selected for metagenomics were from complex bacterial communities (…)”

Round 2

Reviewer 1 Report

The number of replication for donors was not enough. The experiment design was not strict.

Author Response

Comments Reviewer 1 (round 2)

The number of replication for donors was not enough. The experiment design was not strict.

We would respectfully remind the reviewer that the objective of our study was to identify novel lactate utilizers from bovine rumen, and that we were able to successfully achieve this objective. This summary from Reviewer 1 (round 1) indicates that the reviewer is in agreement with our assessment:

“The authors used 16S rRNA gene-based identification of two bacterial OTUs, Bt-01708_Bf (89.0% identical to Butyrivibrio fibrisolvens) and Bt-01899_Ap (95.3% identical to Anaerococcus prevotii), and analyses the predicted proteomes from metagenomics-assembled contigs assigned to these candidates ruminal bacterial species, and revealed genes encoding lactate dehydrogenase, a putative lactate transporter, as well as pathways for the production of short-chain fatty acids (formate, acetate and butyrate), and the synthesis of glycogen. The present study would contribute to the continued characterization of ruminal bacterial species which can metabolize lactate.”

In rebuttal to this reviewer comment, we respectfully disagree that using more donors would have been necessary, because it would not have changed the outcome of the study described in this manuscript. In other words, even if we had used more donors, we would have identified the same candidate lactate utilizers by using this approach. However, if we had reported that lactate utilizers could not be identified, then we would agree that more donors would have been necessary to prove this point, but it is not the case here.

Furthermore, we respectfully disagree that ‘the experiment design was not strict’. Our results show that 1) each candidate lactate-utilizing OTU was found in higher abundance in lactate-supplemented cultures compared to the inoculum or non-supplemented culture, and that 2) assembled contigs assigned to the OTUs encoded for enzymes and proteins that would be required to utilize lactate.

Reviewer 2 Report

The authors have done an excellent job responding to the reviewer's concerns, and have further improved an excellent manuscript.

Author Response

Comments Reviewer 2 (round 2)

The authors have done an excellent job responding to the reviewer's concerns, and have further improved an excellent manuscript.

Thank you again for your kind and very positive assessment of the manuscript. We truly appreciate the recommendations from your previous report as they have strengthen our manuscript.
